# Inferring assembly-curving trends of bacterial micro-compartment shell hexamers from crystal structure arrangements

**Luis F. Garcia-Alles**[1]*, **Miguel Fuentes-Cabrera**[2], **Gilles Truan**[1], **David Reguera**[3,4]

**1** TBI, Université de Toulouse, CNRS, INRAE, INSA, Toulouse, France, **2** Center for Nanophase Materials Sciences, Oak Ridge National Laboratory, Oak Ridge, Tennessee, United States of America, **3** Departament de Física de la Matèria Condensada, Universitat de Barcelona, Barcelona, Spain, **4** Universitat de Barcelona, Institute of Complex Systems (UBICS), Barcelona, Spain

* lgarciaa@insa-toulouse.fr

**Data Availability Statement:** Excel files listing bending and tilting angles and distance values estimated for individual MD snaphots, as well as

## Abstract

Bacterial microcompartments (BMC) are complex macromolecular assemblies that participate in varied chemical processes in about one fourth of bacterial species. BMC-encapsulated enzymatic activities are segregated from other cell contents by means of semipermeable shells, justifying why BMC are viewed as prototype nano-reactors for biotechnological applications. Herein, we undertook a comparative study of bending propensities of BMC hexamers (BMC-H), the most abundant shell constituents. Published data show that some BMC-H, like β-carboxysomal CcmK, tend to assemble flat whereas other BMC-H often build curved objects. Inspection of available crystal structures presenting BMC-H in tiled arrangements permitted us to identify two major assembly modes with a striking connection with experimental trends. All-atom molecular dynamics (MD) supported that BMC-H bending is triggered robustly only from the arrangement adopted in crystals by BMC-H that experimentally form curved objects, leading to very similar arrangements to those found in structures of recomposed BMC shells. Simulations on triplets of planar-behaving hexamers, which were previously reconfigured to comply with such organization, confirmed that bending propensity is mostly defined by the precise lateral positioning of hexamers, rather than by BMC-H identity. Finally, an interfacial lysine was pinpointed as the most decisive residue in controlling PduA spontaneous curvature. Globally, results presented herein should contribute to improve our understanding of the variable mechanisms of biogenesis characterized for BMC, and of possible strategies to regulate BMC size and shape.

## Author summary

Bacterial microcompartments are complex macromolecular ensembles that participate in varied metabolic processes in many microorganisms. They consist of a proteinaceous shell that encapsulates enzymatic cargo that mediate connected chemical reactions. Being confined within shells, the overall efficiency of the process is thought to augment, and reactions that imply toxic intermediates, which are lethal to the host in free-diffusing

the YASARA scripts and input files that would be required to reproduce MD runs are available from the Zenodo database (10.5281/zenodo.7752896). All other relevant data are in the manuscript and its supporting information files.

**Funding:** The French ANR supported financially this work: ANR-19-CE09-0032-01 to LFG-A. The Spanish MICINN is also acknowledged for funding D.R. work (projects PGC2018-098373-B-I00 and PID2021-126570NB-I00). The funders had no role in study design, data collection and analysis, decision to publish, or preparation of the manuscript.

**Competing interests:** The authors have declared that no competing interests exist.

context, become feasible. These natural nano-reactors are therefore appealing for biotechnology purposes.

Understanding how such complex objects form is essential. Experimental studies demonstrated the existence of two possible pathways. Thus, shell components grew onto a preformed core of organized cargo in cyanobacterial carboxysomes, whereas in Pdu compartments, both cargo and shell formation seemed to occur randomly. With the aim to contribute to the understanding of the different biogenesis pathways, here we investigated the trend of the most abundant protein constituents of bacterial micro-compartment shells to form curved structures. We first discovered the occurrence of two major assembly modes in deposited BMC-H crystal structures that clustered proteins in two groups that correlate well to their experimental propensities to form bent or flat structures. Subsequently, molecular dynamic simulations supported that only one of the modes is ready to curve. Moreover, simulations on protein mutants pinpointed a residue that seems to be pivotal in triggering bending. Globally, our data permit to draw a scenario that explains BMC biogenesis differences as a result of the capture of some BMC-H, such as CcmK of carboxysomes, in local minima corresponding to flat states that would delay closure of BMC shells, something that might require the intervention of other molecular effectors.

## Introduction

Bacterial micro-compartments (BMC) are complex macromolecular protein assemblies that confine specialized enzymatic activities within shells and participate in processes like carbon fixation in cyanobacteria or metabolite degradation related to bacterial growth and pathogenesis [1,2]. According to a recent genomic survey, about 70 BMC types can be found in nature, all differing by sort of cargo contents [3]. In contrast, all shell protomers belong to two structural families. The first one adopts the Pfam 00936 fold, which associate as hexamers (BMC-H) or trimers of bidomain proteins (BMC-T). These proteins can be further sub-classified depending on structural details such as for instance the permutation of secondary elements or the capacity to further dimerize [4,5]. Members of the second family (Pfam 03319) assemble as pentamers (BMC-P) and occupy shell vertices [6]. Accordingly, BMC-P are stoichiometrically very minor in shells.

The structure of reconstituted minimalist shells proved that BMC-H are endowed with a considerable assembly versatility, being capable of establishing contacts with themselves in at least two different configurations (planar or bent), but also with BMC-T and BMC-P shell partners [6]. Noteworthy, the same set of residues basically ensured interactions with all different partners, irrespective of the precise local symmetry environment, something that is reminiscent of viral capsids, where a single protein often occupies different structural environments. Yet, according to atomic-force microscopy (AFM) nano-indentation experiments, BMC shells are considerably less rigid than capsids or encapsulins [7].

Despite impressive advances in the structural characterization of BMC, the understanding of shell assembly is progressing slowly. A major milestone was the characterization of the biogenesis of β-carboxysomes from *Synechococcus elongatus* PCC 7942 (*Syn7942*) [8,9]. Colocalization experiments with fluorescently-labelled RuBisCO and shell proteins evidenced that β-carboxysome assembly is a two-step process, shells appearing only after the emergence of preorganized procarboxysome "grains" of coalesced RuBisCO, carbonic anhydrase and scaffolding proteins like CcmM and CcmN (*cargo-first* mechanism, see below). The presence of less

organized/compact encapsulated contents in other BMC types, when compared to hexadeca-meric RuBisCO [10], and the observation of compartment formation in the absence of cognate cargo [11,12], pointed to the existence of other assembly pathways for other BMC types. In line with this, a recent study on Pdu BMC biogenesis proved that *cargo-first* and *shell-first* assembly pathways are both feasible, even when cargo-enzymes coalesce following a hierarchi-cal organization mode [13].

A compelling understanding of assembly is being provided by theoretical simulations using coarse-grained (CG) models to describe simplified cocktails of shell components, cargo pro-teins and even scaffolding factors [14,15]. These studies indicated that assembly pathway, as well as the morphology and cargo-loading extent will be function of the precise balance of interaction strengths between the components and of their stoichiometry. For instance, strong scaffold-mediated cargo-cargo interactions would favour two-step mechanisms, whereas weaker interactions would lead to concomitant scaffold-cargo coalescence and shell assembly. These studies also highlighted that, albeit BMC size is relatively insensitive to hexamer-hex-amer or scaffold-hexamer affinities, it was critically impacted by the assignment of spontane-ous shell curvature and scaffold length [15]. Curvature would result from an imbalance of attractive and repulsive forces established above and below the planes of each interacting pair of hexamers.

Although expected to be critical in driving BMC closure, studies of spontaneous curvature are scarce. To the best of our knowledge, only the assembly of PduA [16] and its interaction with the PduN pentamer [17] were investigated with this intention. In the first study, all-atom molecular dynamics (MD) performed on a pair of interacting hexamers revealed a preference to remain planar. Coarse-grained simulations were subsequently parametrized with 25˚ curva-ture between hexamers, a choice based on angles observed in the cryo-electron microscopy (EM) structure of *Hal. ochraceum* (HO) BMC shells [6]. In contrast, potential of mean force (PMF) calculations presented in the second study revealed a preference for 34˚ inclinations between PduA hexamers. Globally, it remains unclear whether BMC-H are endowed with properties required to induce spontaneous shell curvature or not.

In this study we compiled experimental evidences that support the existence of two major BMC-H assembly classes, depending on preferences to form curved or flat objects, and the possibility that such behavior could be inferred from the kind of organization adopted in crys-tal structures exhibiting piled planar subunit arrangements. All-atom MD performed on tri-hexamer ensembles extracted from these structures globally reproduced experimental trends. BMC-H that display a first organization mode in crystals bent rapidly and reliably towards conformations compatible with the formation of objects characterized with purified proteins [18], closely resembling arrangements observed in minimalist reconstituted BMC shells too [6,19]. On the contrary, hexamers adopting other tiled organizations were reluctant to bend, something that, however, was restored when the same hexamers were repositioned as in the first assembly mode. Globally, our data point to the existence of different structural assembly strategies in BMC-H, with proteins from BMC represented by *β*-carboxysomes being tran-siently trapped in arrangements unsuitable for shell closure.

## Results

### Two general BMC-H assembly behaviors

A considerable effort has been dedicated over the last 20 years to characterize the assembly properties of BMC shells components. S1 Table presents a compilation of conclusions drawn from studies of individual BMC-H, mostly performed by transmission electron microscopy (TEM) and atomic force microscopy (AFM). Apart from highlighting an extraordinary

plasticity, these data served to cluster BMC-H depending on recurrent trends to assemble giving rise to objects exhibiting different bending degrees. A first group of proteins comprise PduA$^{Sent}$, CsoS1A$^{Hneap}$, RMM-H$^{Msm}$ and possibly PduJ$^{Sent}$, which are prone to build rounded structures (nanotubes or spheroids). On the contrary, $\beta$-carboxysome CcmK proteins, possibly BMC-H$^{Hoch}$ too, would more easily organize as (quasi)flat assemblies. Less clear-cut cases, like EutM, seem to fluctuate between bent and flat organizations, depending on organism origin and experimental conditions.

## Different 2D-assembly modes identified in BMC-H crystals

Our intention was to investigate BMC-H assembly behavior by MD simulations taking advantage of crystallographic data. By the time of the realization of this study, there existed about 60 BMC-H structures deposited in the Protein Databank (plus 8 entries from reconstructed shells published in the course of this work). Thirty-four structures were from wild-type (WT) proteins. Within this group, we focused our work on 16 crystal structures that displayed hexamers organized as piled 2D layers (S2 Table, see below).

These 16 structures could be categorized in four groups, which differed by lateral displacements and distances between interacting hexamers (Fig 1 and S2 Table). The first arrangement (hereafter called *Arr-A*) is characterized by a short distance between the two key Lys residues from interacting hexamers. A representative case is the PduA$^{Sent}$ 3NGK structure, with measured 7.2–8.4 Å distances between Lys26 Cα from hexamer counterparts (Fig 1A). *Arr-A* is the only assembly mode observed for WT versions of proteins PduA$^{Sent}$, CsoS1A$^{Hneap}$, CsoS1C$^{Hneap}$ and BMC-H$^{Ahyd}$ (Fig 1B). A second defining feature of this organization mode is the insertion of the Arg79 sidechain (PduA$^{Sent}$ numbering) in an extended conformation within a small pocket between two α-helices of the facing hexamer. Among residues that build this pocket, the presence of a carboxylate (Glu83 in PduA$^{Sent}$) seems to be common to all members of this group of proteins. It is important to highlight that *Arr-A* reflects closely the arrangement of BMC-H noticed in all reconstituted shells (S2 Table). The second assembly mode (*Arr-B*) is adopted by all CcmK proteins, also by BMC-H$^{Hoch}$ and EutM$^{Ecol}$ (Fig 1C and 1D). In this organization, hexamers are more loosely packed (70–71 Å hexamer spacing) than in *Arr-A* (68–69 Å). The inter-lysine distance is considerably longer (14.9–17.6 Å). The side-chains of Arg79-corresponding residues adopt varied conformations, depending on PDB entry, something that seems justified by the presence of neutral residues replacing the Glu83 in the abovementioned R79-binding pocket of PduA$^{Sent}$. This structural difference, combined to the *Arr-B*-specific ionic interaction established between residues corresponding to R28-D49 of CcmK1$^{6803}$ (R28 being replaced by Asn/Glu in *Arr-A* BMC-H) might together contribute to assemble BMC-H differently (Fig 1D). An intermediate organization (*Arr-C*), with distances of about 10 Å between lysines, is observed in CcmK2$^{6803}$ 3DNC and CcmK4$^{7942}$ 4OX6 structures, whereas CcmK2$^{7942}$ 4OX7 is the only case displaying a fourth assembly type (*Arr-D*).

Arrangement occurrence did not seem to be defined by crystallization conditions. Thus, the organization mode was reproduced in crystals of the same protein prepared under variable conditions. For instance, *Arr-A* is adopted in structures 2EWH and 2G13 from CsoS1A$^{Hneap}$, or in the 3H8Y structure of the close CsoS1C$^{Hneap}$ homolog. Similarly, CcmK1$^{6803}$ was characterized with *Arr-B* arrangement, both in 3BN4 and 3DN9 structures, likewise 3MPW and 3MPY structures from EutM$^{Ecol}$. A single protein, CcmK2$^{6803}$, was found to happen in two different packings: *Arr-B* (2A1B, 3CIM) and *Arr-C* (3DNC).

Globally, a concordance was evidenced between the way a given BMC-H tiles in crystals and the experimental assembly behavior. Thus, *Arr-A* would be the preferred crystallization mode for proteins endowed with curving propensity, whereas flat-behaving BMC-H would

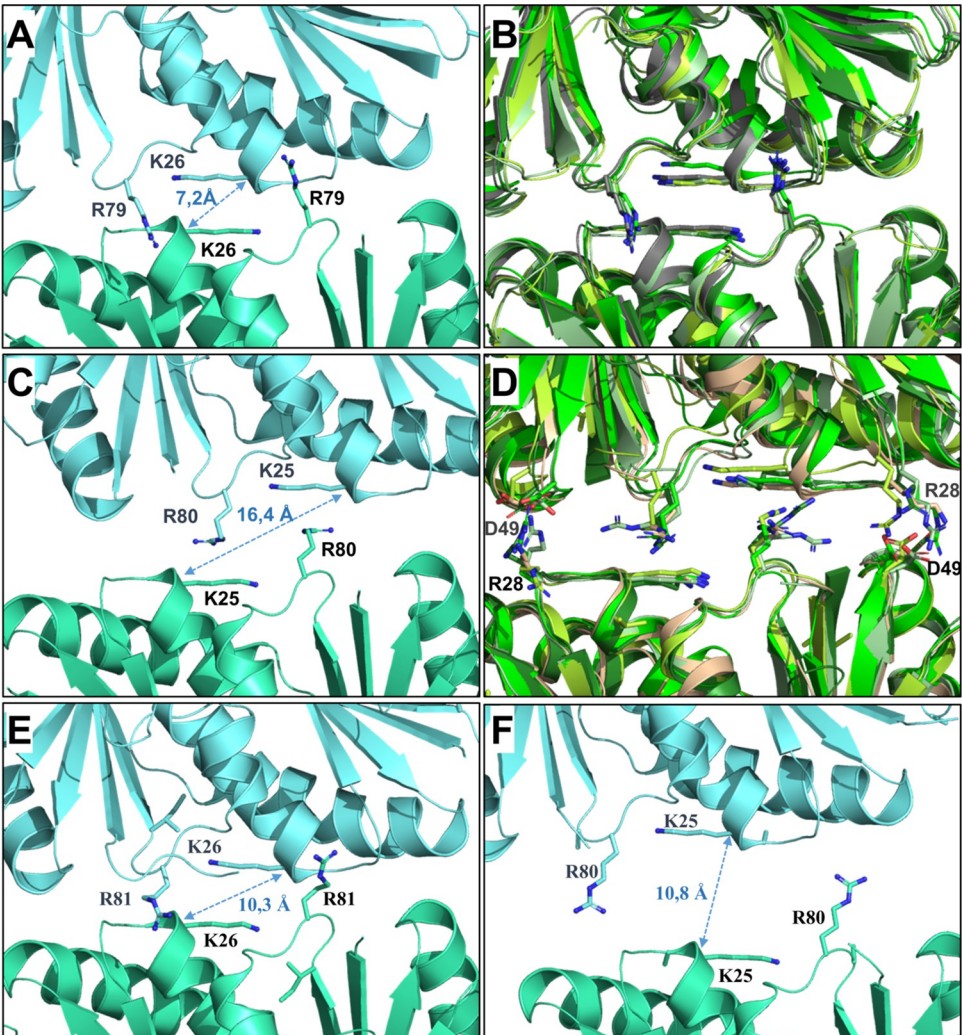

**Fig 1. Two major 2D-arrangements in crystals of BMC-H.** The two most abundant organizations occurring in crystals of 2D-tilling BMC-H are shown in panels *A-D. Panels A* and *C* correspond to PduA$^{Sent}$ (3NGK) and CcmK1$^{6803}$ (3BN4) structures, which adopt *Arr-A* and *Arr-B* organizations, respectively. The close-up view of the inter-hexamer interface is shown with hexamers in ribbon representation colored cyan or blue marine. Side-chains of key Lys and Arg are shown as sticks, with nitrogen atoms in deep blue. Distances are measured between alpha carbons of the two Lys; In *panels B* and *D* are shown PDB entries with BMC-H arranged similarly to PduA (3NGK) or CcmK1$^{6803}$ (3BN4), respectively, which are colored in grey for comparison. Identity of BMC-H in *panel B*: in green PduJ (5D6V) with restored K25, in pale green CsoS1A (2G13), in limon CsoS1A (3H8Y) and BMC-H$^{Ahyd}$ (4QIV) in forest green; in *panel D*: CcmK1$^{6803}$ (3BN4) in green, CcmK2$^{6803}$ (2A1B) in pale green, CcmK4$^{6803}$ (6SCR) in limon, EutM$^{Ecol}$ (3MPW) in forest green and BMC-H$^{Hoch}$ (5DJB) in wheat. These views were generated after super-imposition of structures based on backbone atoms from the bottom hexamer; Ionic interactions between Arg28 and Asp49 of CcmK1$^{6803}$, or corresponding residues, are established in *Arr-B* organizations. *Panels E,F* provide views of unusual assembly types *Arr-C* [CcmK4$^{7942}$ (4OX6)] and *Arr-D* [CcmK2$^{7942}$ (4OX7)], respectively.

mostly adopt *Arr-B* arrangements. However, some BMC-H like EutM would not obey such trends, displaying intermediate properties between PduA and CcmK cases.

## Variable bending trends supported by all-atom molecular dynamics

The hypothetical relationship between organization in crystals and spontaneous curvature was tested by means of all-atom MD. We followed the approach described in a previous study [20],

in which MD runs were launched on ensembles of three interacting hexamers, extracted from crystal structures showing tiled BMC-H (PDB codes indicated in S3 Table). Tri-hexamers were selected as good compromise to describe the situation in BMC shells while keeping reasonable computational costs. MD were run (Amber ff14SB force field) after energy minimization of initial models with proteins embedded in explicit hydration boxes under quasi-physiological conditions (see M&M). All different structural organizations mentioned in the previous section were covered. Hexamer tilting and bending angles were monitored for intermediate structures extracted in the course of each trajectory (250 ps snapshots), as well as inter-hexamer distances calculated from the hexamers center of mass in the MD average structure. Rather than long single simulations, we opted by performing several independent simulations on each case (20 ns each), which differed by the attribution of random initial atom velocities. This parameter could impact more profoundly than time length the MD trajectories of rapid processes like those studied here [21], something that we indeed confirmed (see below).

MD behavior was defined by the kind of arrangement adopted in the crystal structure. Thus, strong and reproducible bending was noticed for all *Arr-A*-deriving cases. This is illustrated for PduA$^{Sent}$ (3NGK) in Fig 2, but similar trajectories were recorded in simulations with triplets of hexamers deriving from 2EHW, 3H8Y, 3NGK, 5D6V and 4QIV PDB entries (Fig 3). For all these cases, practically all bending angles estimated for the many MD snapshots were negative and distributed within a narrow interval of values (Fig 3A, see also S1 Fig and S3 Table). Remarkably, average bending values of *ca* -25˚ were reproduced in 4 independent runs with PduA$^{Sent}$ and CsoS1A$^{Hneap}$ (2G13). The less pronounced effect occurred for PduJ$^{Sent}$. Although we believe this likely reflects the intrinsic lower curvature propensity of this BMC-H, it is necessary to mention that the corresponding tri-hexamer was prepared from the K25A mutant 5D6V entry, which was manually modified to re-introduce native interfacial K25 side-chains.

As expected, bending was accompanied by a slight decrease of inter-hexamer separations (not to be confused with edge to edge inter-hexamer distance) (S3 Table). Induction of curvature was rapid, reaching poses close to the MD average during the first nanosecond (Fig 2C). Consequently, data dispersion for all cases was low, even smaller than those measured for EutL$^{Ecol}$, a BMC-T that basically remained flat in two MD runs. A simultaneous evolution of angles for the three different hexamer pairs of the tri-hexamer was noticed in the beginning of some MD runs, suggesting the occurrence of coordinated movements. Importantly, negative bending corresponded to the orientation described for full BMC shells. Indeed, root-mean-square deviations (RMSD) of only 1.3 Å were measured (on 1392 backbone atoms) when comparing interacting pairs of monomers in the average structure of two independent MD on PduA$^{Sent}$ (3NGK) with corresponding interacting monomers of bent BMC-H in the *H. ochraceum* BMC shell structure (5V74). Worth-mentioning, the results were reproduced using CHARMM as force field in similar independent simulations launched on PduA$^{Sent}$ (average bending angle of -16 ± 6˚) and CsoS1A$^{Hneap}$ (-22 ± 6˚).

Assembly fate was more uncertain for non-*Arr-A* organizations. Both bending and tilting were much more variable depending on the snapshot, and often distributed towards the two possible orientations, giving rise to much stronger data dispersion (Fig 3). Overall, the reproducibility between runs was also considerably poorer. An exception was the *Arr-C* case CcmK2$^{6803}$ (3DNC), which evolved much like *Arr-A* assemblies, although trajectories were characterized by a higher data dispersion. Sometimes, average structures were almost flat in different runs [e.g. CcmK1$^{6803}$ (3BN4) or CcmK4$^{7942}$], but for most other cases trajectories were instable, with simulations that even conducted to averages with positive sign bending [e.g. with EutM$^{Ecol}$ or CcmK2$^{6803}$ (2A1B)]. Despite the somehow chaotic behaviors, globally

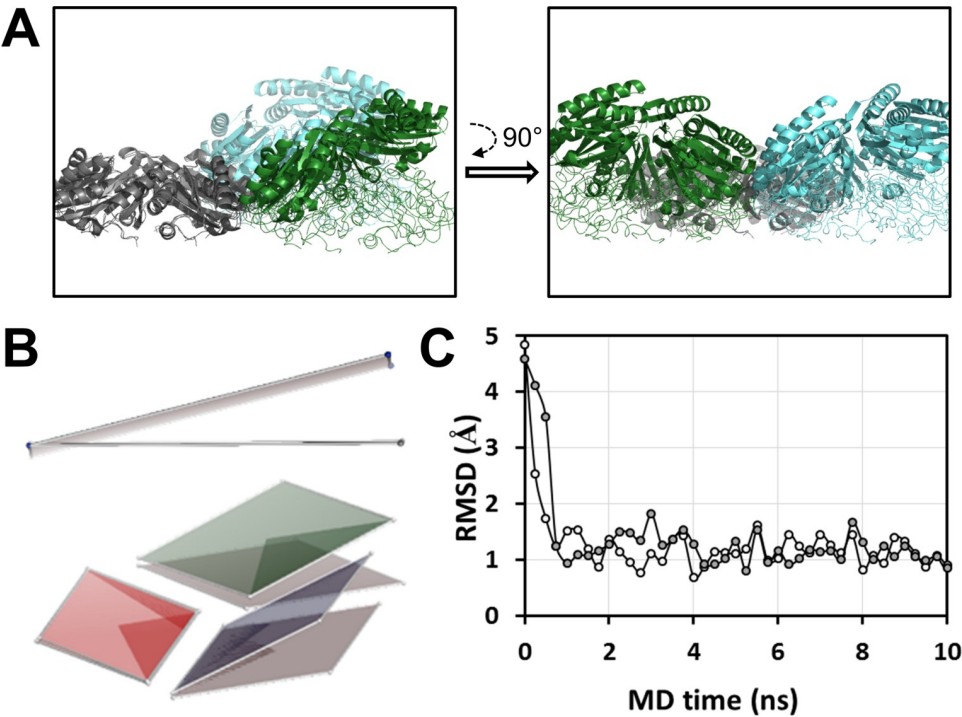

**Fig 2. MD simulations on the PduA$^{Sent}$ tri-hexamer.** *A*, Comparison of the average structure of a 20 ns MD simulation on PduA$^{Sent}$ (cartoon) with the situation at time 0, corresponding to the 3NGK crystal structure (thin traces). The two structures were superposed on backbone atoms of the most to the left hexamer (left view). An orthogonal view is shown in the right. *B*, The same comparison is illustrated by two means: On top, with planes that were elaborated from the coordinates of the center of mass (COM) of each hexamer. Grey spheres are from the starting structure, in blue for the MDs average structure. The view is approximately seen as in the left representation of panel A. Consequently, the traverse view of the plane in the crystal structure results in a flat trace. Bending during the MDs induces the trace to displace upwards or downwards. In the bottom representation, hexamers of the starting structure or MD average are represented by gray or colored planes, respectively. Each plane was prepared from the Cα positions of Ala53 residues from the three monomers of each hexamer that contact other hexamers. *C*, Assembly evolution in the course of PduA$^{Sent}$ MD simulations. Plotted are the RMSD values calculated when the coordinates of backbone atoms from each snapshot structure (0,25ns steps) were compared to the MD average structure (empty circles for first MD, gray for the second run). All structures were superimposed prior to the calculation.

speaking, the angle distributions calculated from MD snapshots for non-*Arr-A*, especially for Arr-B, were interpreted as symptomatic of an absence of bending preference.

## Theoretical behavior of assembled BMC-H substructures extracted from characterized shells

The possibility that *Arr-A* configurations were responsible for BMC-H curving was evaluated in simulations launched on 2 interacting BMC-H extracted from structures of BMC shells (so far, there does not exist any example of tri-hexamers in synthetic shells). As highlighted in S2 Table, BMC-H always adopted *Arr-A*-like organizations in characterized minimalist shell structures. The two hexamer configurations observed for the BMC-H$^{Hoch}$ 5V74 shell were assayed [6], which correspond to either bent or flat starting models, as well as a pair of CcmK$^{7418}$ hexamers with bent orientation taken from the 6OWF shell structure [19].

MDs trajectories clearly showed a preference for curved states: bent BMC-H$^{Hoch}$ or CcmK$^{7418}$ remained close to the starting angle, while the planar BMC-H$^{Hoch}$ rapidly evolved towards a curved configuration (S2 Fig).

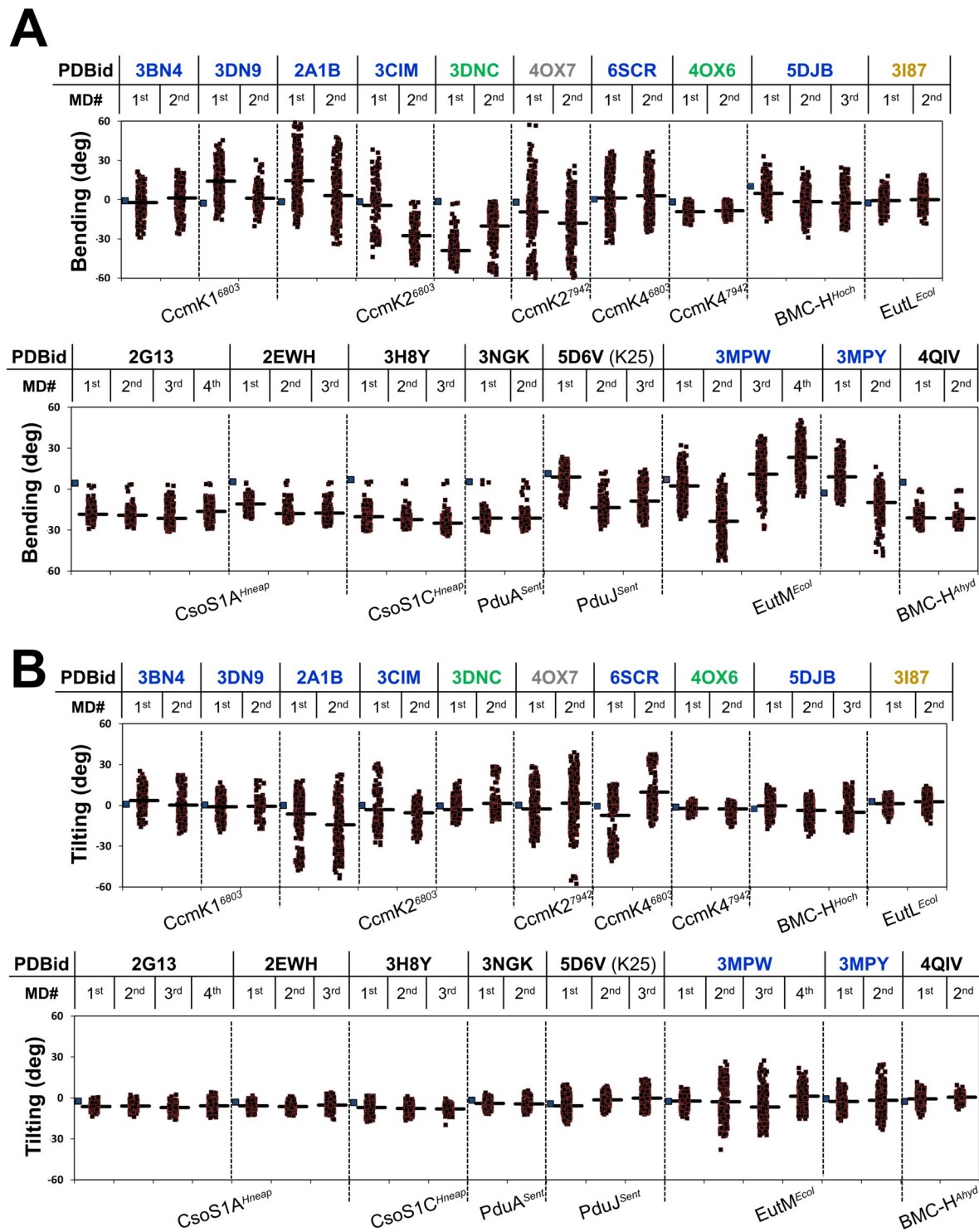

**Fig 3. BMC-H tri-hexamer behavior during MD simulations.** Plot of bending (panel *A*) and tilting (*B*) angles calculated through all-atom MD trajectories of ensembles of three BMC-H originally positioned as in crystal structures (indicated by PDB entry codes on top). Bending angles were calculated taking the Cα atom positions of S27 and Ile38 from one of the interfacial monomers of a given hexamer (PduA$^{Sent}$, corresponding residues in other BMC-H) with regard to same residues on the symmetric monomer of the interacting hexamer counterpart. That was applied on all 3 hex-hex interfaces. Similarly, tilting values are based on Cα positions of M24 and Ile18 from two monomers of a given hexamer with regard to the same residues on the symmetric monomers of the contacting hexamer. Each point corresponds to one of three measurements for a given snapshot (0.25 ns, black squares). Data from up to 4 independent 20 ns simulations are presented separately (1$^{st}$ to 4$^{th}$). Thick traces represent the mean value calculated over the entire MD run. Blue squares on the most left side for each PDB entry

give the angle values measured for the original crystal structure. Occasionally, readings might be impacted by local distortions of protein backbone (see S1 Fig for alternative views of MD averages). PDB codes are colored according to the type of organization: black for *Arr-A* arrangements, blue for *Arr-B*, green for *Arr-C* and grey for *Arr-D*. For comparison, results obtained on the EutL$^{Ecol}$ BMC-T (3I87) are presented in the last two columns of the upper portion of each panel.

## Atomic determinants triggering BMC-H bending

Most challenging aim was the identification of atomic determinants implied in triggering curvature. We selected for such study PduA$^{Sent}$, because of its demonstrated experimental trend to form nanotubes and robust MD behavior. First, we sought to establish key interactors that clamp hexamers together. Two analytical approaches were followed: i) side-chain RMSD with regard to the MD average structure were monitored over the MD run (S3 and S4 Figs). For each residue, the different 18 positions in the tri-hexamer were plotted together. A clamping residue was expected to result in a relatively fixed conformation, and thus in lower RMSD, but only when located at the interface. In that manner, Lys26 and Arg79 always occurred with lowest RMSD at interfaces, for two independent MDs. Interfacial Glu19, Asp22, Asn29, Pro78 and His81 were often, but not always, lowest; ii) the contribution to the interaction energy of each residue was evaluated. The snapshot with lowest RMSD to the average structure of each MD was selected and energy-minimized. We considered as contributors to the interaction those residues that resulted in maximal interval of values, when comparing values for the 18 monomers (S5A Fig), and at the same time presented highest stabilization when located at the interface, when compared to exposed positions (S5B Fig). In that manner, Lys26, Arg79 were again confirmed to be key contributors, together with Ser27. The importance of Lys26 and Arg79, but not of Ser27, was confirmed when the contribution of each residue to the potential energy was averaged over the snapshots of a full MD trajectory (S5C Fig). The only additional important contributor identified in that manner was Asn29.

The role of above-pinpointed residues for assembly fate was investigated by *in silico* MD of single-residue alanine mutants. We extended the exploration to other interfacial residues to anticipate unforeseen possibilities, also to positions known to establish contacts between hexamers in characterized BMC shells, such as Asn67, which corresponds to Arg66 of CcmK$^{7418}$ [19]. In total, the next 21 residues were scanned: K12, E19, D22, K26, S27, N29, R48, D50, V51, K55, D59, R66, N67, H75, P78, R79, H81, T82, D83, E85 and K86 (mutations were introduced 6 or 9 times in the tri-hexamer, depending on whether the residue was close to the center or edge of the interface, respectively). Notably, curvature was only impacted by the K26A mutation (S6 Fig), which completely abolished bending and even led to 1–2 Å larger hexamer separation. The result was reproduced in four independent 20 ns MDs (only two shown). All other mutations were without effect, including the R79A. None of 6 side-chains tested replacing the interfacial K26 was able to restore bending (S7 Fig), a result that agrees with the full conservation of this residue among BMC-H [4]. Interestingly, bending of WT PduA$^{Sent}$ was impeded when simulations were run with artificial neutral charges imposed to the sidechains of the 6 interfacial K26 residues. Calculated average bending values for the first and second MD were -6.8 (± 5) and -8.7 (± 8) degrees, respectively. Overall, these simulations evidence a major role of positive charges on the ammonium groups of K26 residues of PduA in sensing the overall electrostatics and triggering displacements from the flat and tight *Arr-A* assembly towards curved states.

We next evaluated the universality of such effect in other BMC-H. The importance of the corresponding lysine, also of Arg79, the two residues suspected to play the most critical role at the interface, was estimated in the same manner as with PduA$^{Sent}$. A moderate reduction of bending was noticed for the K25A BMC-H$^{Ahyd}$ (S8 Fig), the extent of the effect being variable

among MD runs. On the contrary, the R78A mutation was again without consequence. Similarly, the CsoS1A$^{Hneap}$ behavior was not perturbed by either K29A or R83A mutations. Accordingly, the role played by K26 of PduA$^{Sent}$ should not be generalized to other BMC-H.

## Multiple energy minima in lateral contacts between planar BMC-H

Our MD results, together with the structures of all recomposed shells (S2 Table), concur to prove that *Arr-A* is the ready-to-curve configuration. That most other tiling BMC-H adopted a second organization (*Arr-B*) could therefore rule out casual coincidences, and instead argue in favor of the possibility that *Arr-B* could reflect a structural trap that would delay shell closure. To verify this hypothesis, two approaches were envisioned. First, we evaluated the interaction energy profile after gradually displacing the relative lateral localization of hexamers. Umbrella sampling all-atom MD [22] were performed on hexamer couples extracted from PduA$^{Sent}$ (3NGK), CcmK1$^{6803}$ (3BN4) or CcmK4$^{7942}$ (4OX6) structures, taken as representative of *Arr-A*, *Arr-B*, and *Arr-C* organizations, respectively (Fig 4). Preliminarily, we measured the PMF that results from pulling apart the two partners (Fig 4A). Despite comparable to calculations on CcmK2$^{6803}$ [23], CcmK1$^{6803}$ binding energy was very weak, about 2 to 4 times smaller than values estimated for CcmK4 or PduA, respectively. When the PMF was screened along the order parameter that leads from *Arr-A* to *Arr-B* (further extended on both sides), while constraining hexamers to remain flat, the highest stability was attained around crystal positions, supporting a preference of PduA$^{Sent}$ for *Arr-A*, or of CcmK4$^{7942}$ for *Arr-B/C* arrangements (Fig 4B). CcmK1$^{6803}$ did not demonstrate any clear preference.

Measured profiles did not permit to conclude convincingly on whether arrangements occurring in 2D-tiling crystals represent global energy minima or not, especially for cases adopting *Arr-B* assemblies. Nevertheless, and although energetic differences will be most likely

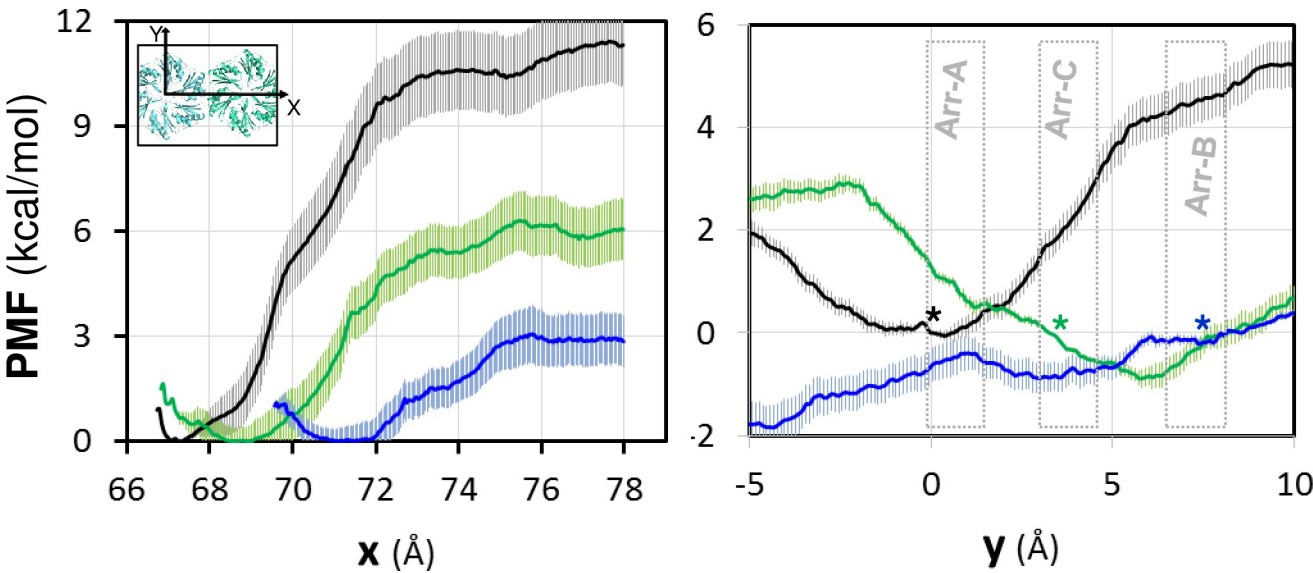

**Fig 4. Potential of mean force (PMF) between two BMC-H hexamers.** In the left panel, the PMF was calculated using umbrella sampling all-atom MD simulations after separating progressively the two hexamers along the x-axis defined by the two center of masses (COM) of the hexamers in the starting X-ray structure (inset). The origin of energy is taken at the minimum of the PMF so that the value of the energy at the largest distance provides an estimation of binding energies. For the right panel, the hexamers were gradually displaced along the orthogonal y-axis and the PMF was calculated using restraints to prevent bending, tilting and z-rotation. Data for PduA$^{Sent}$ (3NGK) is plotted in black, in blue for CcmK1$^{6803}$ (3BN4) and green for CcmK4$^{7942}$ (4OX6), including error bars estimated by bootstrapping. The approximate location of the different assembly modes is indicated in the right panel. The asterisks are to indicate the approximate position of the starting crystal for each case (following the mentioned color code).

amplified within the context of more realistic extended ensembles, shallow profiles revealed for BMC-H interactor couples fitted with the perception of BMC-H interfaces being endowed with strong structural plasticity. These results also support that transitions between different assembly states should be feasible.

### Behavior of reconfigured assemblies

In the second approach, MDs simulations were launched on tri-hexamers of CcmK, EutM$^{Ecol}$ and BMC-H$^{Hoch}$ (remaining flat in crystals), after artificially repositioning each hexamer in an *Arr-A* configuration. For that, each hexamer was superimposed individually on the different hexamers of the template PduA$^{Sent}$ (3NGK) and potential clashes conveniently relaxed taking as template the conformation of PduA residues (see M&M). We decided to include RMM-H$^{Msm}$ in the study, in view of its demonstrated potential to form nanotubes [24].

MD launched on reconfigured assemblies revealed significantly more stable than when starting from crystal layouts, and collectively confirmed that *Arr-A* likely represents an arrangement competent for shell closure for most, if not all BMC-H (S9 Fig). Thus, BMC-H$^{Hoch}$ and RMM-H$^{Msm}$ behaved much like CsoS1A$^{Hneap}$ or PduA$^{Sent}$ (S1 Fig). Also remarkable was the CcmK4$^{7942}$ curving trend, contrasting with the reproducible flatness of this protein when arranged as in the 4OX6 crystal (S1 and S2 Figs). An exception was CcmK4$^{6803}$, which remained flat. However, the inter-hexamer separation increased by almost 2 Å, similarly to CcmK2$^{7942}$ (S4 Table), something that could point to insufficiently relaxed starting structures. Since several bulky residues lie at the junction of the three CcmK4$^{6803}$ hexamers and might hamper bending, we assessed a mutant with several residues replaced by corresponding residues from PduA$^{Sent}$: R30N, Q53G, E54A, E85T and N86D. This mutant bent, albeit still less pronouncedly than all other cases (S9 Fig).

### Discussion

Understanding how macromolecular structures as complex as BMC or BMC shells form is challenging. Implying the controlled assembly of hundreds to thousands of oligomeric subunits [25] deriving from several components, such processes are thought to rely in cooperative phenomena. Relatively weak interactions, in the low $k_B$T regime, normally govern contacts between pairs of most abundant shell components [23]. That explains why coarse-grained models are precious tools to investigate parameters governing these processes [14,15], in spite of being unsuitable for predictive work aiming to engineer new assemblies.

With the intention to contribute to this effort, this study sought to investigate global relationships between experimental assembly behavior of BMC-H and predictions from all-atom MD simulations. Compiled experimental data pointed to a segregation of BMC-H in two major groups (S1 Table), depending on trends to form flat assemblies (basically *β*-carboxysome CcmK) or rounded objects (e.g. Pdu or CsoS1 proteins). Within the first group could be included cases like BMC-H$^{Hoch}$, found here or before to assemble flat *in vitro* [26,27], and leading to rosette structures inside *E. coli* that are assumed to derive from spooling of quasi-flat carpets [26,28,29]. Nanotubes/spheroids and planar structures are not necessarily mutually exclusive, as demonstrate for example published data for EutM homologs assembled at variable temperatures [30]. Transitions between flat and curved structures might be one of the consequences of the relatively shallow interaction energy profiles calculated here or before for BMC-H [16,17,23].

A notable discovery was that BMC-H experimental assembly behavior appeared to be related to the type of organization found in crystals that exhibit internal 2D-layered organizations. Bending cases like PduA$^{Sent}$, PduJ$^{Sent}$ or CsoS1A$^{Hneap}$ exhibited *Arr-A* structuration,

whilst other arrangements were noticed in structures of CcmK, EutM$^{Ecol}$, EutM$^{Cdif}$ or BMC-H$^{Hoch}$. Most often, assembly units were laterally displaced by approximately 8–10 Å (*Arr-B*), when compared to *Arr-A*. If such relationship was correct, crystal data would indicate that proteins like CsoS1C$^{Hneap}$, CsoS1$^{Pmar}$ and BMC-H$^{Ahyd}$ should form rounded structures, temptingly also BMC-H that attained *Arr-A* states even when mutated: CmcB$^{Ecol}$ (7MN4, K25A-E55G mutant), CmcC$^{Ecol}$ (7MPX, K25A-E35G) or CutR$^{Sint}$ (6XPI, K66A). However, this rule is not obeyed by all BMC-H, since EutM is prone to form rounded structures but adopts *Arr-B* configurations in crystals. EutM complex behavior is illustrated by the varied types of structures formed depending on the homolog selected [30].

That hexamers could organize differently within 2D-layered crystals was already known since the early report of CsoS1A$^{Hneap}$ crystal structure [31]. At that time, side-to-side hexamer packing differences between CsoS1A (66.4 Å inter-hexamer separation) and CcmK structures (69.7 or 70.5 Å for CcmK1$^{6803}$ or CcmK4$^{6803}$, respectively) were argued to reflect a means to attain compatibility among shell constituents or a mechanism to regulate the porosity of the shell. The availability of many more structures nowadays rules out that differences were coincidental or induced by crystallization conditions, which spread considerably by pH (4.2 to 9.5) or additives within each group (S2 Table), and occasionally overlapped between the two classes (compare for instance conditions with 4OX8 vs 4OX6). Indeed, PMF profiles estimated here support that the highest stabilization is attained in different flat arrangements of PduA$^{Sent}$ and CcmK4$^{7942}$ hexamers. Although CcmK1$^{6803}$ profile was unexpectedly shallow, that any CcmK ever reached an *Arr-A* configuration in crystals (10 PDB entries) should be taken as indicative of global minima different from *Arr-A*. Yet, the structures of all reconstituted BMC shells strongly suggest that *Arr-A* is the mode of organization that conducts to closed assemblies (S2 Table), also in shells containing CcmK proteins [19].

All-atom MDs supported the experimental curving capabilities of *Arr-A*-organized BMC-H. Not only PduA$^{Sent}$, but also all other *Arr-A* BMC-H rapidly and robustly bent. The known bending orientation was reproduced, i.e. the convex face lying towards the compartment lumen. In fact, structures averaged over the MD were strikingly similar to those found in minimalist shells, which all appear to derive from *Arr-A* arrangements. Comparison with data published for PduA$^{Sent}$ bi-hexamers [16] indicate that assembly might be better simulated on tri-hexamers, although in our hands a preference for curved structures was even demonstrated on bi-hexamers extracted from minimalist BMC-H$^{Hoch}$ and CcmK$^{7418}$ shells. In fact, more recent PMF calculations from the same authors on PduA$^{Sent}$ bi-hexamers also pointed to slightly more stable bent states than flat organizations [17]. BMC-H with non-*Arr-A* organizations behaved less clear-cut, bending and tilting values being strongly dispersed over the MD trajectories. Globally, the system curved towards the two sides of the tri-hexamer plane, generating dispersed distributions of values measured for the different snapshots throughout the MD. This was seen for individual MD runs [e.g. CcmK4$^{7942}$ (4OX6)], or was concluded from independent MDs [e.g. CcmK2$^{6803}$ behavior in assemblies prepared from crystals 2A1B and 3CIM]. Conversely, reproducible trajectories were recorded for these BMC-H, when re-arranged to reproduce an *Arr-A* state, with bending always occurring towards the convex face (S9 Fig). We therefore interpreted these data as indication of a potential mechanism to prevent premature BMC closure in *β*-carboxysomes, a process that would require a transition between assembly modes and might require the participation of other molecular effectors (see below).

Unveiling atomic determinants behind spontaneous curvature was expected to be especially challenging, since these ensembles are supposed to rely on cooperative effects. Difficulties are illustrated by the striking failure to experimentally disrupt BMC-H assemblies when key residues are mutated. This was the case of K26A, V51A and R79A PduA$^{Sent}$ mutants [18], of K28A and R78A BMC-H$^{Hoch}$ [26,29], or of R28A, D49A and R80A CcmK1$^{6803}$ [20]. Structured

objects continued to form, in spite of the fact that interactions between constituting subunits are indeed weak. In view of such difficulties, it was somehow unexpected that the mutation of the interfacial K26 (PduA) completely and reproducibly abolished bending *in silico*. However, the effect did not prevail for other BMC-H, although the residue is fully conserved. Also striking, none of the other 20 PduA$^{Sent}$ mutants tested modified the behavior. From our point of view, MD results globally suggested that PduA bending could be defined by the electrostatics around the K26 ammonium group. Energetic terms applied around this charge could abruptly shift when transiting towards *Arr-A* tight packings. Indeed, a structural chart illustrating the localization of charged residues lying near the interface in different BMC-H suggests a potential source of electrostatic imbalance, as most of ionizable residues that lie close to the hexamer-hexamer interface reside on the concave face (S10 Fig). Accordingly, stronger repulsion forces might be exerted around the K26 ammonium originating from that side. The relaxation of such forces, articulated in the context of other interactions that would clamp the hexamers together (see below) would result in bending.

Energetic calculations proved that Arg79 is also a key assembly interactor (S5 Fig). An important role in controlling curvature was attributed to this residue recently [16]. In that study, the mutation of PduA R79 in a ΔPduJ strain was shown to hamper the formation of nanotubes or even of Pdu BMC. Yet, mutation of other positions had already been reported to be of similar consequences [e.g. for K26A PduA [18] or K25A PduJ [32]], inviting caution before concluding on cause/effect relationships. In our hands, R79A did not exert any significant effect on curvature *in silico*. Supporting that K26 could play a more decisive role for bending, Pdu compartments were not recovered from *S. enterica* when chromosomal PduA or PduJ were replaced by their respective K26A or K25A mutants, whereas compartments still formed with the R79A mutant and the protein was even co-purified with the BMC [33]. Piled 2D sheets were also imaged by TEM with a K26A mutant of a variant of PduA from *Cit. freundii* expressed in *E. coli* [18]. However, in this last study the R79A mutation elicited a similar result. Accordingly, the two residues might be proposed to act concertedly. In fact, K26 sidechain is often modeled in crystals with a stretched conformation, lying antiparallel with regard to the same residue of the counter-interacting hexamer. Asp22, also fully conserved, contributes to hold such conformation. Consequently, the positively-charged groups of K26 and R79 are brought closer, something that might impose an extended conformation to the R79 sidechain. In that manner, the insertion of this sidechain in a small pocket of the facing hexamer, observed in all *Arr-A* structures, might be facilitated, an advantage that would be lost in the K26A (and R79A) mutant. In support of this scenario, a carboxylate group present in the mentioned pocket of PduA$^{Sent}$ (Asp83) might anchor R79 side-chains of the hexamer counterpart. Coincidently, Asp or Glu residues occur at the same position of all *Arr-A* BMC-H, whereas a neutral Asn is found in CcmK proteins and in BMC-H$^{Hoch}$. Yet, Asp83 did not seem to contribute to stabilization of the bent tri-hexamer (S5C Fig). Regardless of Asp83 implication, an R79A mutation would perturb the assembly process, and indirectly bending if attainment of the *Arr-A* state was perturbed. However, such effect would be out of reach for our short MD simulations, which started already with *Arr-A* assembled tri-hexamers. Ongoing experimentation should permit to verify the importance of this ionic interaction in driving *Arr-A* formation, and thus bending, and similarly, whether the replacement by the R28-D49 ionic pair contributes to trap planar proteins in *Arr-B* arrangements. Interestingly, sequence alignments indicate that EutM$^{Ecol}$ would be the only case among studied cases that would combine the two ionic pairs, something that might fit to its more complex behavior. Importantly, the conservation of residue types within paralogs indicates that such assembly mechanisms would even be preserved in the case of formation of hetero-hexamers [34,35].

Irrespective of the mechanisms that trigger curvature, our observations have implications for the interpretation of BMC biogenesis. *β*-carboxysome formation was proven to be a two-step process in *Syn. elongatus* PCC 7942, the presence of pre-organized cargo preceding shell assembly [8,9]. A similar study on Pdu*^Sent^* compartments pointed instead to a less synchronized mechanism, with coexistence of *cargo-first* and *shell-first* events [13]. A comparable model could operate with α-carboxysomes, as indicate the gathering of CsoS2 scaffolding protein to portions of shells preceding recruitment of RuBisCO and final BMC closure [36], or the multiple layers of RuBisCO attached to the inner surface of partial α-carboxysome shells [10,37]. In view of that, and considering that BMC-H are the most abundant shell components, *Arr-B* might reveal a structural trap that would prevent premature shell closure of some BMC such as *β*-carboxysomes. Transit towards *Arr-A* configurations would be triggered by contacts with motifs on cargo/scaffold components or, alternatively, upon intervention of auxiliary proteins such as CcmO or CcmP. Indeed, BMC-T co-expression was required to induce the formation of *Halothece* sp. PCC 7418 or BMC-H*^Hoch^* empty shells, both inside cells [19,38] or *in vitro* [39], while BMC-T presence was dispensable for the formation of minimalist α-carboxysomes [40] or *Klebsiella pneumoniae* GRM2 shells [41]. In the last study, CmcC from GRM2 was proposed to be a component endowed with high spontaneous curvature, in agreement with the crystal *Arr-A* organization of the *E. coli* K25A CmcC homolog [42]. Future studies are necessary to investigate the potential implication of BMC-T in mediating BMC-H assembly transitions, also to establish the physiological consequences that would have the alteration of the natural biogenesis pathway of a given BMC.

## Materials and methods

### All-atom molecular dynamics simulations

Assemblies composed of three hexamers were prepared from available structures (PDB ID indicated in S2 Table) after applying crystallographic translation and symmetry operations. Glycerol and other crystallographic ligands were removed (sulfate ions associated to CsoS1A were deleted, or not, without evident difference). Hexamer triplets were embedded in cuboid cells with dimensions extending 20 Å around all protein atoms, which were filled with explicit solvent. Each system was neutralized with NaCl (0.9% final concentration). Periodic boundary conditions were applied and, unless otherwise mentioned, the YASARA Amber14 (ff14SB) force field was selected. The cut-off for the Lennard-Jones potential and the short-range electrostatics was 8 Å. Long-range electrostatics were calculated using the Particle Mesh Ewald (PME) method with a grid spacing <0.1 nm, 4th order PME-spline, and PME tolerance of $10^{-5}$ for the direct space sum. YASARA's pKa utility was used to assign pKa values at pH 7.0. The entire system was energy-minimized using steepest descent minimization, followed by a simulated annealing minimization until convergence (<0.05 kJ/mol/200 steps). The equations of motions were integrated with a multiple time step of 2.5 fs for bonded interactions and 5.0 fs for non-bonded interactions at a temperature of 298K and a pressure of 1 atm (NPT ensemble). At least two simulations were launched on each case, with attribution of random initial atomic velocities (20 ns/run, unless otherwise indicated). Intermediate MD snapshots were recorded every 250 ps. Simulations were carried out in a 16-core CPU PC exploiting GPU capabilities (NVIDIA GeForce GTX 1080), and lasted typically 50 to 60 hours per 20 ns run.

For simulations of *Arr-A* reconstituted assemblies, hexamers extracted from indicated PDB structures were superimposed individually on the different hexamers of the template PduA*^Sent^* (3NGK) tri-hexamer. For most cases, sterical clashes around R30, the D51-E54 segment and the R82-N86 region (CcmK4*^6803^* numbering) were alleviated by adapting the side-chain conformation to reproduce those present in the corresponding residue of PduA (3NGK). Special

attention was given to the CcmK conserved Arg30. Its side-chain conformation was adapted to reproduce the orientation observed for Arg28 in the cryo-EM structure of the CcmK[7418] shell (6OWF). In CcmK4[6803], the Arg30, Gln53 and Glu54 collapse close to the C3 axes of symmetry of the tri-hexamer. These side-chains were therefore adapted manually. The resulting models were in all cases thoroughly minimized, before launching MD simulations that also included a minimization phase, as explained before.

Dihedral angle and distance analysis was performed with adapted Python scripts executed under Pymol (https://www.pymol.org/). Tilting was measured among $C\alpha$ atom positions of M24 and Ile18 (PduA[Sent], corresponding residues in other BMC-H) from two monomers of a given hexamer with regard to same symmetric residues of the interacting hexamer counterpart, as described [20]. Bending values were based on $C\alpha$ atoms of S27 and Ile38 from one of the interfacial monomers and the same residues on the symmetric monomer of the opposite hexamer. For plane representations, structures averaged over the MD were first superimposed on the corresponding crystal structure. Only backbone atoms from one of the hexamers was used for the superimposition. Next, each hexamer was represented by the plane containing V54 $C\alpha$ atom (PduA[Sent]) of the three monomers of each hexamer that enter in contact with other hexamers at the interface. In other representations, each hexamer was represented by its center of mass, calculated considering only backbone atoms of core residues (res 1 to 90) from the six monomers. Structural figures were also prepared with Pymol.

Data from MD trajectories snapshots, either in YASARA.sim format or as.pdb files are available upon request.

## Umbrella sampling molecular dynamics simulations

Assemblies of two hexamers of PduA[Sent] (3NGK), CcmK1[6803] (3BN4), and CcmK4[7942] (4OX6) were prepared at pH 7.0 from the available crystal structure using ProteinPrepare (https://playmolecule.com/proteinPrepare/). All atom MD simulations were performed using GROMACS (version 2021.1), with Amber ff99SB-ILDN force field. All other conditions were as mentioned above, with the difference that the cut-off for the Lennard-Jones potential and the short-range electrostatics was established at 10 Å. The entire system was energy-minimized using steepest descent minimization, followed by a constant volume, constant temperature (NVT) equilibration at T = 298 K for 100 ps with the backbone of both hexamers restrained with a force constant of 1000 kJ/mol/nm$^2$.

Umbrella sampling simulations were performed with translation and rotation of the first hexamer prevented with harmonic restraints (force constant of 1000 kJ/mol/nm$^2$) applied to six $C_\alpha$ atoms (one per monomer, near its COM). Binding energies were evaluated using the distance between the two hexamers COM as order parameter. A steered simulation pulling on the second hexamer with a force constant of 1000 kJ/mol/nm$^2$ at a rate of 10 nm/ns was used to generate configurations at 1 Å steps along the order parameter. A total of 15 umbrella windows per case were therefore simulated to reconstruct each PMF. In each window a harmonic potential with a spring constant k = 1000 kJ/mol/nm$^2$ was used as a biasing potential. The biasing potential was centered on different values of the order parameter equispaced at intervals of 1 Å. For each window, a NPT simulation of 2 ns was performed for equilibration, followed by a 10 ns NPT simulation sampled every 50 ps for producing the histogram that is then used to get the PMF between the two hexamers using the Weighted Histogram Analysis Method (WHAM) [22]. Equilibration in each window was corroborated by monitoring the convergence of the potential energy, typically reached well before the first nanosecond. The error was estimated using bootstrapping. The PMF as a function of the hexamer lateral displacement as order parameter was calculated similarly, with the difference that the second hexamer was

progressively displaced using UCFS Chimera from the crystal position, at 1 Å steps in a range between -15 Å and + 15 Å. In these simulations the bending, tilting, and z-rotation angles were restrained to zero with harmonic potentials with a force constant 1000 kJ/mol/rad$^2$ to keep the two hexamers planar.

## Supporting information

**S1 Table. Nano-assemblies characterized for individual BMC-H.** Data were compiled from references indicated in the last column. Objects were imaged by TEM directly after protein overexpression inside living cells (generally *E. coli*), or by TEM, cryo-EM or AFM with purified proteins (*in vitro*). Curved-implying objects are highlighted with blue letters, black for flat structures. NA: not applicable.
(DOCX)

**S2 Table. Assembly types in crystal structures with tiling BMC-H.** In total, 54 structures of BMC-H were extracted from the RCSB databank and inspected. Only hits presenting planar arrangements of hexamers or 1D-stripped organizations were retained. Wild-type BMC-H entries studied herein by MD simulations are indicated in black bold letter. Non studied cases appear in grey and include data from mutants. A few other cases presenting planar arrangements were excluded from the table because contacts were either too weak or did not follow canonical arrangements: PduA$^{Sent}$ (4RBT), PduU$^{Sent}$ (3CGI), CmcB$^{Ecol}$ (7MPW), CutN$^{Sint}$ (7MMX), CcmK2$^{Telo}$ (3SSR) or BMC-H$^{Hoch}$ (6NLU). The 5$^{th}$ and 6$^{th}$ column report distances measured between alpha carbons of either Lys26 or Arg79 (PduA$^{Sent}$) interfacial residues from interacting hexamers (corresponding residues in other BMC-H or mutants). The second portion of the table provides information on BMC-H organizations present in structures from reconstituted BMC. Only the first one (5V74) was obtained by crystallographic methods, all other by cryo-EM.
(XLSX)

**S3 Table. Structural changes of tri-hexamers assemblies occurring during MDs trajectories.** [a] Tilting and bending values correspond to the average of deviations measured between each MD snapshot structure and the crystal structure. The averages combine the three measurements between each couple of hexamers in the tri-hexamer assembly. For bending angles, negative sign indicates orientation towards BMC-H convex side. In occasions, local structural distortions around residues selected for calculation of angles could result in misleading values. Angles therefore need to be contrasted with plane representations prepared taking the center of mass (COM) of hexamers (see S1 Fig). [b] Deviation of distances were calculated taking the coordinates of the COM of each hexamer in the MD average structure with regard to the crystal. Only main-chain atom coordinates of residues 1–90 in each monomer were considered for the estimation of each hexamer COM. The value is the average of the three inter-hexamer measurements. [c] The *Aver* column provides the mean value of all independent MD runs. [d] Values corresponding to MD runs carried out with CHARMM forcefield. [e] For PduJ$^{Sent}$ (5D6V), the alanine mutated residue in position 26 was replaced by the lysine residue of the wild-type protein. NP: not possible, as calculation of COM position was hampered by monomer dislocations through the simulation box walls, which occurred during the run. PDB id codes are colored according to assembly type, likewise in S1 Fig.
(DOCX)

**S4 Table. Structural changes during MDs trajectories when the starting tri-hexamer is reconfigured by positioning individual BMC-H hexamers as in *Arr-A* mode.** PDB codes are colored according to the type of organization in original structures that served to prepare the

starting model for MD simulations (see S9 Fig for further details). For other details on how measurements were carried out, please refer to S3 Table. Please notice that local structural distortions might occur around residues selected for calculation of angles, which could result in misleading angle values. Such alterations might be amplified in the context of reconfigured interfaces. Angles therefore need to be contrasted with plane representations prepared taking the center of mass (COM) of hexamers (S9 Fig).

(DOCX)

**S1 Fig. Comparison of average structures from MD simulations on BMC tri-hexamer ensembles with starting crystal structure.** Two type of illustrations are presented on the left or right sides. They were prepared following the scheme explained in Fig 2B. PDB codes are colored according to the type of organization: black for *Arr-A* arrangements, blue for *Arr-B*, green for *Arr-C* and grey for *Arr-D*.

(TIF)

**S2 Fig. MD simulations on bi-hexamers from BMC shells.** *A*, All-atom molecular dynamics of an ensemble of two hexamers extracted from published structures (PDBid codes indicated on top) and positioned in planar or curved configurations depending on the selected shell environment. See Fig 2 for further details. *B*, Comparison of structures generated by averaging atom positions over the MD snapshots (cartoons) with the starting structure (thin traces). The two structures were superposed on backbone atom coordinates of one of the hexamers (shown in cyan, on the left side). From top to bottom: bent BMC-H$^{Hoch}$ (5V74), flat BMC-H$^{Hoch}$ (5V74) and bent CcmK$^{7418}$ (6OWF).

(TIF)

**S3 Fig. RMSD evolution of PduA residues over MD simulations.** Represented is the average of root-mean-square deviations (RMSD) measured between side-chain atoms of indicated residues in each MD snapshot when compared to the residue atom coordinates in the structure averaged over the first PduA MD run. The first 8 snapshots were excluded from the calculations. Each panel present the values for a given residue in each of the 18 monomers of the tri-hexamer. Values in monomers from the first hexamer are shown in black to light grey scale, from the second hexamer with blue tonalities, green for the third. The arrows are to indicate residues from monomers that enter in contact with a neighboring hexamer. Similar results were obtained from data collected in the second MD run. Residues outlined in continuous red systematically show lower RMSD when placed at the inter-hexamer interface, in independent MD runs. Discontinuous outlines are for those residues that occur often, but not always, with lower RMSD.

(TIF)

**S4 Fig. Sidechain movements of selected PduA residues during MD simulations.** The view presents the side-chain conformations adopted by the several residues in the different collected snapshots of the first MD run on PduA$^{Sent}$, depending on whether the residue lies at the inter-hexamer interface (left panels) or not (right): *A-B*: Glu19; *C-D*: Asp22; *E-F*: Lys26; *G-H*: Asn29; *I-J*: Arg79. Side-chains are represented as sticks, with nitrogens blue and oxygens in red. Residues were selected from data presented in S3 Fig. All snapshot structures were superimposed on main-chain atoms of one of the hexamers (black cartoon). The two other hexamers are shown in blue or green traces.

(TIF)

**S5 Fig. Energetic contribution of PduA residues to the stabilization of the bent assembly.** *A*, Interval of energies contributed by every residue of PduA, when comparing the 18

monomers of the tri-hexamer assembly. The ordinate presents the energy interval measured between the less and most stabilizing position. The most similar (lowest RMSD) snapshot to the averaged structure of the first MD run was selected for the analysis and energy computation was done with GROMOS96 implemented in Swiss-PDBViewer. *B*, Estimated energy contribution of selected residues in the 18 different emplacements of the trihexamer. Only a few residues among those analyzed are presented. Data are colored as in S3 Fig. The arrows are to identify residues in monomers that lie at the interface with neighbor hexamers. Outlined in red are residues that contribute a stabilizing effect when positioned at the inter-hexamer interface, for both MD runs. Similar results were obtained from data collected in the second MD run. *C*, Lys26, Asn29 and Arg79 were identified as stabilizing residues from plots of potential energy contributions averaged over the snapshots that covered the entire MD run. Here, potential energies were calculated using Amber (ff14SB) forcefield implemented in Yasara, and a single MD run was analyzed.
(TIF)

**S6 Fig. MD behavior of PduA tri-hexamer assemblies with selected residues mutated into alanine.** *A*, Residues indicated in the first lane were replaced by alanine in the 6 or 9 monomers of the trihexamer assembly that lie at the interface. An assembly with only three K26 positions mutated was also simulated. The Cα of such residues is indicated by red spheres in the second column. The result of two MD runs is presented following plane representations like those of Fig 3. Indicated bending angle values were evaluated like in S3 Table. Please notice that local distortions caused during the MD might in occasions result in under/overestimations. In the case of the K26A mutant (6x, outlined in red), four MD runs were carried out, with similar qualitative results. *B*, Effect of mutation on the distance between hexamers during the MD run. In the ordinate axes is represented the difference between the averaged distance calculated for the three hexamers (center of masses) in the averaged structure of a given MD simulation, and the average distance calculated from four independent MD runs on wild-type (WT) PduA, which are shown on the most left side, following the value measured for the PduA crytal (3NGK). Data obtained in independent MD run repetitions are denoted by 1$^{st}$ and 2$^{nd}$ label extensions below the X-axis.
(TIF)

**S7 Fig. MD consequences of replacement of K26 of PduA by other residue types.** Lys26 was replaced by residues indicated in the first column in the 6 monomers located at the contacting interface between subunits. The result of two MD runs is presented following plane representations explained in Fig 2B (right side).
(TIF)

**S8 Fig. MD behavior of *Arr-A* BMC-H tri-hexamers with interfacial Lys and Arg mutated into alanine.** Key interfacial Lys and Arg, residues indicated in the second column, were replaced in CsoS1A$^{Hneap}$ (2G13) or BMC-H$^{Ahyd}$ (4QIV) by alanine in the 6 monomers of the tri-hexamer assembly that lie at the interface. Data from several independent MD runs are presented. Data for wild-type versions are shown for the ease of comparison. Hexamers are represented in the two ways presented in Fig 2B. Please notice that indicated average bending angles, calculated as explained in S3 Table, might be impacted by local displacements around the main-chain atom positions selected for the measurements.
(TIF)

**S9 Fig. Dynamic behavior of tri-hexamers reconfigured in Arr-A arrangement.** Comparison of the average structure generated for all snapshots of each MD simulation with the structure at time 0. The starting tri-hexamer ensemble was prepared by superimposition of an

individual hexamer taken from the indicated PDB entry on each different hexamer of a template PduA$^{Sent}$ (3NGK) trihexamer, followed by manual interventions and energetic relaxation to prevent residue clashes (see M&M). Representations were prepared likewise in Fig 2B. PDB codes are colored according to the type of organization in the original PDB: blue for *Arr-B*, green for *Arr-C* and grey for *Arr-D*. Please note that RMM was included in the study, but the 5L38 entry do not belong to cases with tiled arrangements of hexamers. The CcmK4$^{6803}$ pentamutant carried the next changes with regard to the WT version: R30N (6x), Q53G (9x), E54A (9x), E85T (6x) and N86D (6x).
(TIF)

**S10 Fig. Unbalanced distribution of ionic residues in the 3D structures of BMC-H.** Represented is the localization of Cα atoms of cationic (Arg, Lys and His, blue spheres) or anionic residues (Asp, Glu, in red) around the interface of two hexamers. Only ionic residues lying closer than 7 Å from any atom of the hexamer counterpart are represented. Portions of each hexamer are depicted with green or grey traces, views being generated along the interface axis (left side of each panel, with concave side being at the bottom), or from top of the hexamers as seeing from the convex side (right). Left *A-D* panels correspond to *Arr-A* 3D structures: *A*, PduA$^{Sent}$ (3NGK); *B*, K25 PduJ$^{Sent}$ (5D6V); *C*, CsoS1A$^{Hneap}$ (2G13); *D*, BMC-H$^{Ahyd}$ (4QIV). Right panels are from *Arr-B* organized proteins, after reconfiguration of hexamers in *Arr-A* mode for the ease of comparison: *E*, CcmK1$^{6803}$ (3BN4); *F*, CcmK4$^{6803}$ (6SCR); *G*, BMC-H$^{Hoch}$ (5DJB); *H*, EutM$^{Ecol}$ (3MPW).
(TIF)

**S1 File. README file explaining contents of supplementary files deposited in the Zenodo database.**
(DOCX)

## Acknowledgments

CHARMM simulations by M.F.-C. were performed at the Center for Nanophase Materials Sciences, which is a US Department of Energy Office of Science User Facility at Oak Ridge National Laboratory.

## Author Contributions

**Conceptualization:** Luis F. Garcia-Alles.

**Data curation:** Luis F. Garcia-Alles, David Reguera.

**Formal analysis:** Luis F. Garcia-Alles, David Reguera.

**Funding acquisition:** Luis F. Garcia-Alles.

**Investigation:** Luis F. Garcia-Alles, Miguel Fuentes-Cabrera, David Reguera.

**Methodology:** Luis F. Garcia-Alles, David Reguera.

**Project administration:** Luis F. Garcia-Alles.

**Supervision:** Luis F. Garcia-Alles.

**Validation:** Luis F. Garcia-Alles.

**Visualization:** Luis F. Garcia-Alles.

**Writing – original draft:** Luis F. Garcia-Alles.

**Writing – review & editing:** Gilles Truan, David Reguera.

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
