## [Decision Letter · Decision Letter 0]

4 Mar 2023

Dear PhD Garcia-Alles,

Thank you very much for submitting your manuscript "Inferring assembly-curving trends of bacterial micro-compartment shell hexamers from crystal structure arrangements" for consideration at PLOS Computational Biology. As with all papers reviewed by the journal, your manuscript was reviewed by members of the editorial board and by several independent reviewers. The reviewers appreciated the attention to an important topic. Based on the reviews, we are likely to accept this manuscript for publication, providing that you modify the manuscript according to the review recommendations.

Sincerely,

David van der Spoel

Academic Editor

PLOS Computational Biology

Arne Elofsson

Section Editor

PLOS Computational Biology

Reviewer's Responses to Questions

**Comments to the Authors:**

Reviewer #1: This is an interesting study of assembly modes of a wide range of different BMC hexamers. The authors use several different all-atom MD techniques to try and understand the spontaneous curvature of BMC assemblies, which is difficult given the computational expense. Some of the results especially for the well-studied PduA have been seen before, but in my opinion, reproduction of previous results only strengthens the reliability of the methods used and is quite useful for the community. Furthermore, this study is quite impressive in the number of different BMC hexamers which were studied including many point mutations. Overall, the conclusions are somewhat limited by the inability to measure energetic differences corresponding to different bending states as they can with different flat configurations, but it is still a good study. It should be published after addressing the following points.

Specifics:

Line 98: The authors state only the curvature of PduA has been investigated with respect to closure. I would modify that statement slightly as Mills, et al. Nat Commun 13, 3746 (2022) investigated the role of PduN (which is a pentamer) and its interaction with PduA in the context of shell closure, measuring intrinsic curvature in the interactions.

Line 167, tri-hexamer section: My understanding is that the bending angle distributions include all 3 measured bending angles from the 3 interfaces in the tri-hexamer configuration. I am wondering if there are correlations between the 3 bending angles which were measured simultaneously for the 3 different pair interfaces in 3 hexamer configurations. In other words, does knowing the bending angle of one interface tell you what the angle of the other 2 interfaces? I assume they must be somehow correlated based on geometry. Can the authors comment on this somewhere in this section?

Line 213, bi-hexamer section: This goes along with the previous point, but do you expect to see the same bending angles in a bi-hexamer as a tri-hexamer configuration? You mention cooperative effects in the manuscript and choose tri-hexamer configurations where possible (as opposed to the less computationally expensive bi-hexamer) so I am wondering if you believe it would be appropriate to assume the same bending angles would be found in a tri-hexamer configuration. Was such a hypothesis tested here or previously? You mention even on line 171 the tri-hexamer is a compromise. Can you comment on this in this section?

Fig 4: The Potential of Mean Force is labeled as E, but this normally corresponds to an enthalpy or internal energy, whereas the authors calculated a free energy, which is a more useful quantity. I think this label should be changed from E to F or PMF to make it clear that this is a free energy calculation as opposed to just measuring energies in a simulation as in Figure S5.

Reviewer #2: The authors did a decent job in addressing the reviewer's questions, suggestions, and concerns.

MINOR ISSUES:

L 179/180: Referring to running several short simulations versus a single long one, the authors write: "This parameter is known to impact MD trajectories more profoundly than time length (20), something that we indeed confirmed (see below)." This statement is not generally true, but it depends on whether one has an activated process (with an activation barrier), which leads to a rare event that, when it happens, is in itself fast (but the system has along dwell time in the initial minimum), or whether the process is in itself slow. The latter can be, for example, a slow phase transition or sth that is related to mass transport/diffusion over long distances, etc. In this case case, one obviously really needs long simulations. In the former case, running several short simulations can indeed be better (and is trivially parallelisable on a HPC machine).

L 200 (and several places elsewhere in the text): There are different AMBER and CHARMM force field parameter sets. So it would be helpful it the authors would always specify which one exactly they refer to. Concerning the statement in L 200 that the results were reproduced with the CHARMM force field, are these data shown anywhere (in SI)? If not, this will probably have to be removed.

Some wording is strange here and there. One example is in L 210: "distribution of snapshot angles"? Obviously, there is no angle between snapshots, so this sentence has a logic problem. What is meant is the angle distributions calculated from the MD configurations (snapshots). Maybe the authors could carefully check the text for such things. In addition, the word "disposition" is still used at several places in the text. "Disposition" simply means sth else in English than what the authors intend to say, so perhaps correct this everywhere.

Reviewer #3: The authors have fully addressed my comments and made all requested changes, and I feel that the manuscript is ready for publication. I do have one additional suggestion though, that the authors include the plot of the average energies (in their response to my first "main point") in the Supporting Information of the manuscript so that readers can compare the single-configuration energies to the average energies.

**Have the authors made all data and (if applicable) computational code underlying the findings in their manuscript fully available?**

Reviewer #1: Yes

Reviewer #2: None

Reviewer #3: None

PLOS authors have the option to publish the peer review history of their article (what does this mean?). If published, this will include your full peer review and any attached files.

Reviewer #1: No

Reviewer #2: No

Reviewer #3: No

Figure Files:

Data Requirements:

Reproducibility:

References:

---

## [Editor Report · Decision Letter 1]

21 Mar 2023

Dear PhD Garcia-Alles,

We are pleased to inform you that your manuscript 'Inferring assembly-curving trends of bacterial micro-compartment shell hexamers from crystal structure arrangements' has been provisionally accepted for publication in PLOS Computational Biology.

Best regards,

David van der Spoel

Academic Editor

PLOS Computational Biology

Arne Elofsson

Section Editor

PLOS Computational Biology

---

## [Editor Report · Acceptance letter]

31 Mar 2023

PCOMPBIOL-D-23-00208R1 

Inferring assembly-curving trends of bacterial micro-compartment shell hexamers from crystal structure arrangements

Dear Dr Garcia-Alles,

I am pleased to inform you that your manuscript has been formally accepted for publication in PLOS Computational Biology. Your manuscript is now with our production department and you will be notified of the publication date in due course.

With kind regards,

Anita Estes
